# Rapid deep learning-assisted predictive diagnostics for point-of-care testing

Seungmin Lee[1,2,8], Jeong Soo Park[1,3,8], Hyowon Woo[1,8], Yong Kyoung Yoo[4], Dongho Lee[5], Seok Chung [3], Dae Sung Yoon [2,6,7], Ki-Baek Lee[1] & Jeong Hoon Lee [1,5] ✉

Prominent techniques such as real-time polymerase chain reaction (RT-PCR), enzyme-linked immunosorbent assay (ELISA), and rapid kits are currently being explored to both enhance sensitivity and reduce assay time for diagnostic tests. Existing commercial molecular methods typically take several hours, while immunoassays can range from several hours to tens of minutes. Rapid diagnostics are crucial in Point-of-Care Testing (POCT). We propose an approach that integrates a time-series deep learning architecture and AI-based verification, for the enhanced result analysis of lateral flow assays. This approach is applicable to both infectious diseases and non-infectious biomarkers. In blind tests using clinical samples, our method achieved diagnostic times as short as 2 minutes, exceeding the accuracy of human analysis at 15 minutes. Furthermore, our technique significantly reduces assay time to just 1-2 minutes in the POCT setting. This advancement has the potential to greatly enhance POCT diagnostics, enabling both healthcare professionals and non-experts to make rapid, accurate decisions.

In Point-of-Care Testing (POCT), achieving both high sensitivity and affordable rapid diagnosis is a pivotal challenge. POCT methods are broadly categorized into immunoassay-based and molecular-based approaches. Recent advancements in molecular diagnostics have shown the potential to reduce assay time to less than 10 min using plasmonics and microfluidic techniques[1–3]. However, in the case of most commercialized molecular diagnostics, a sample preparation step is inevitably involved, leading to a relatively lengthy diagnosis time of up to several hours[4–7] (See Supplementary Tables 1, 2).

On the other hand, in immunoassay-based diagnostics, short detection times based on nanosensors, such as nanowires and field-effect transistor (FET) sensors, have been reported[8–10]; however, few have received FDA approval. The commercialized immunoassay platform encompasses enzyme-linked immunosorbent assay (ELISA),

fluorescence Immunoassay (FIA), chemiluminescent immunoassay (CLIA), and lateral flow assay (LFA). ELISA, as the most popular immunoassay platform, requires a significant amount of time, approximately 3–5 h for analysis[11,12]. In contrast, rapid kits, also known as rapid diagnostic tests (RDT), provide quicker results, typically within 15 min, providing the fastest immunoassay[13] (See Supplementary Tables 1, 2).

In the domain of emergency medical care, expeditious and precise diagnosis within the emergency room (ER) holds utmost significance. The patients arriving at the ER often present with severe, life-threatening, or time-sensitive conditions, necessitating prompt and accurate diagnostic interventions[14–16]. For example, cardiac troponin I, which is highly specific to myocardial tissue and undetectable in healthy individuals, is significantly elevated in patients with myocardial

[1]Department of Electrical Engineering, Kwangwoon University, 20 Kwangwoon-ro, Nowon, Seoul 01897, Republic of Korea. [2]School of Biomedical Engineering, Korea University, 145 Anam-ro, Seongbuk, Seoul 02841, Republic of Korea. [3]School of Mechanical Engineering, Korea University, 145 Anam-ro, Seoungbuk-gu, Seoul 02841, Republic of Korea. [4]Department of Electronic Engineering, Catholic Kwandong University, 24, Beomil-ro 579 beon-gil, Gangneung-si, Gangwon-do 25601, Republic of Korea. [5]CALTH Inc., Changeop-ro 54, Seongnam, Gyeonggi 13449, Republic of Korea. [6]Interdisciplinary Program in Precision Public Health, Korea University, Seoul 02841, Republic of Korea. [7]Astrion Inc, Seoul 02841, Republic of Korea. [8]These authors contributed equally: Seungmin Lee , Jeong Soo Park, Hyowon Woo. ✉e-mail: jhlee@kw.ac.kr

infarction and can remain elevated for up to 10 days post-necrosis. Levels above 0.4 ng/ml indicate a notably higher 42-day mortality risk[17]. Particularly for myocardial infarction patients who present to the emergency room, prompt diagnosis and management are crucial. In such critical scenarios, the rapid identification of diseases and conditions exerts a profound impact on patient outcomes.

Notably, in cases involving infectious diseases, timely diagnosis plays a pivotal role in identifying the causative pathogens and infections[18,19], thereby facilitating the timely implementation of infection control measures to avert potential outbreaks[20] and safeguard the health of both patients and healthcare providers[21,22].

Furthermore, for pregnant patients in the ER, knowing their pregnancy status is crucial, especially when considering medical imaging involving radiation[23,24], anesthesia[25,26], or treatments[27] that could affect fetal well-being. Fast and precise diagnosis is key in guiding informed decisions, enabling the effective management of health conditions while simultaneously minimizing risks to both the patient and the fetus.

While LFA is generally recognized as a rapid and commercially viable diagnostic tool[28], its significance in enabling timely interventions extends beyond its immediate applications. LFA also holds a pivotal role in reducing unnecessary tests and treatments, thereby contributing to more efficient healthcare utilization and cost-effectiveness[29,30]. Consequently, the approaches to further shorten assay time while retaining sensitivity have elicited considerable interest, given its potential to unlock numerous novel detection opportunities. These advancements show promise, particularly in emergency medicine, infectious disease management, and neonatal care, with the potential to improve patient outcomes[31–33].

Artificially intelligent (AI) technology has emerged as a focal point in medical image-based diagnostics using convolution neural networks (CNN), encompassing modalities such as X-ray[34,35], computed tomography (CT)[36], and magnetic resonance imaging (MRI)[37,38], with its application promising significant enhancements in diagnostic accuracy while revolutionizing the interpretation and analysis of complex medical images. Recently, our group proposed deep learning-assisted smartphone-based LFA (SMART[AI]-LFA) and demonstrated that integrating clinical sample learning and two-step algorithms enables a cradle-free on-site assay with higher accuracy (>98%)[39]. However, the earlier study primarily highlighted the performance of AI-enhanced colorimetric assays and did not specifically address the reduction of assay time using AI.

Several recent studies in medical diagnostics have emphasized the reduction times by integrating deep learning techniques[40–45]. Previous studies have successfully achieved shorter histopathology tissue staining times using generative adversarial network (GAN)-based virtual staining[40,41] and applied deep learning methodologies to enhance efficiency in plaque assays[42]. Moreover, the utilization of long short term memory (LSTM) deep learning algorithms has expedited polymerase chain reaction (PCR) analysis[43], enabled the prediction of infections based on time-series data from affected individuals[44], and facilitated the utilization of longitudinal MRI images for predicting treatment responses[45]. Meanwhile, the demand for diagnostic tools achieving shortened assay time and maintained sensitivity remains high, but few studies address for achieving AI-assisted fast assay, especially for POCT. Consequently, there is a pressing need for AI technology to enable rapid diagnosis in POCT, representing a transformative step in enhancing diagnostic efficiency beyond traditional hardware optimization.

In this study, we present an approach that combines a time-series deep learning algorithm with lateral flow assay platforms, notably the most affordable and accessible POCT platform, to achieve a significant reduction in assay time, now as short as 1–2 min. Our method, which utilizes an architecture comprising YOLO, CNN-LSTM, and a fully connected (FC) layer, notably accelerates the COVID-19 Ag rapid kit's assay time, facilitated by the Time-Efficient Immunoassay with Smart AI-based Verification (TIMESAVER). This approach is versatile, applicable to a range of conditions including infectious diseases like COVID-19 and Influenza, as well as non-infectious biomarkers such as Troponin I and hCG. In blind tests with clinical samples, our method not only achieved diagnostic times as short as 2 min but also surpassed the accuracy of human analysis traditionally completed in 15 min.

## Results

### Workflow of TIMESAVER for fast assay

Figure 1 presents three representative commercialized diagnostic tools: commercial LFA, PCR, and ELISA, along with their performance in terms of time, labor, cost, and accuracy. Generally, commercial PCR and ELISA tests take several hours, are labor-intensive, and incur higher costs. In contrast, rapid kits typically provide cost-effective, on-site diagnostics. We introduce TIMESAVER-assisted LFA, an approach that combines time-series deep learning architecture, AI-based verification, and enhanced result analysis to optimize LFA immunoassays. Our objective is to establish the fastest diagnostic time among existing commercially available kits while maintaining accuracy and affordability. Conventional rapid kit protocols typically require 10–20 min for analysis, posing challenges in time-sensitive applications like emergency medicine, infectious disease management, neonatal care, and heart stroke, where further assay time reduction is crucial.

As shown in Fig. 1, our approach utilizes a time-series deep learning architecture and AI-based verification, resulting in a significant reduction in assay time to within 1–2 min using TIMESAVER. A more detailed discussion of the time-series deep learning architecture, known as the TIMESAVER algorithm, is provided in Fig. 2. This algorithm is specifically designed for learning from time-series data and has effectively reduced diagnosis times. Notably, the results demonstrate diagnosis times as short as 1–2 min for LFA when utilizing a smartphone or reader (See Supplementary Movie 1).

### Model optimization for TIMESAVER algorithm

In Fig. 2, we present the deep learning architecture, TIMESAVER, utilized for predicting results, which consists of three components: YOLO, CNN-LSTM, and the FC layer. Figure 2a illustrates the overall scheme of TIMESAVER, a deep learning architecture consisting of three interconnected components. This involves transforming the entire image into a cropped image containing the test line, which is then processed through CNN and LSTM networks to generate a vector representation. Subsequently, the CNN and LSTM outputs are combined and passed through the FC layer to produce the predicted result.

Region of Interest (ROI) selection is a crucial step in rapid kit diagnosis (Fig. 2b). The selection of the Region of Interest (ROI) enhances the accuracy of detecting the specific concentration of the target biomarker or pathogen, thereby increasing sensitivity and specificity and minimizing the occurrence of false negatives and false positives[39]. As detailed in our previous research, we investigated two methods for ROI selection in LFAs: focusing on the window and the test line exclusively. The approach centered on the window area achieved a prediction accuracy of 92.9%, while a focus exclusively on the test line enhanced the prediction accuracy to 95.2%. Data augmentation is a vital technique, particularly for limited or imbalanced datasets (Fig. 2c). It involves applying various transformations to existing data, generating synthetic images to enrich the dataset and enhance the model's robustness. In our study, we acquired RGB channel images and transformed them into HSV channel images. The data augmentation results were as follows: RGB achieved an accuracy of 95.2%, HSV achieved 64.3%, and combining RGB and HSV yielded a perfect accuracy of 97.6%.

In Fig. 2d, we optimized the CNN model. For feature extraction from images, we used a CNN specifically designed for image recognition and processing tasks, making CNNs essential in computer vision

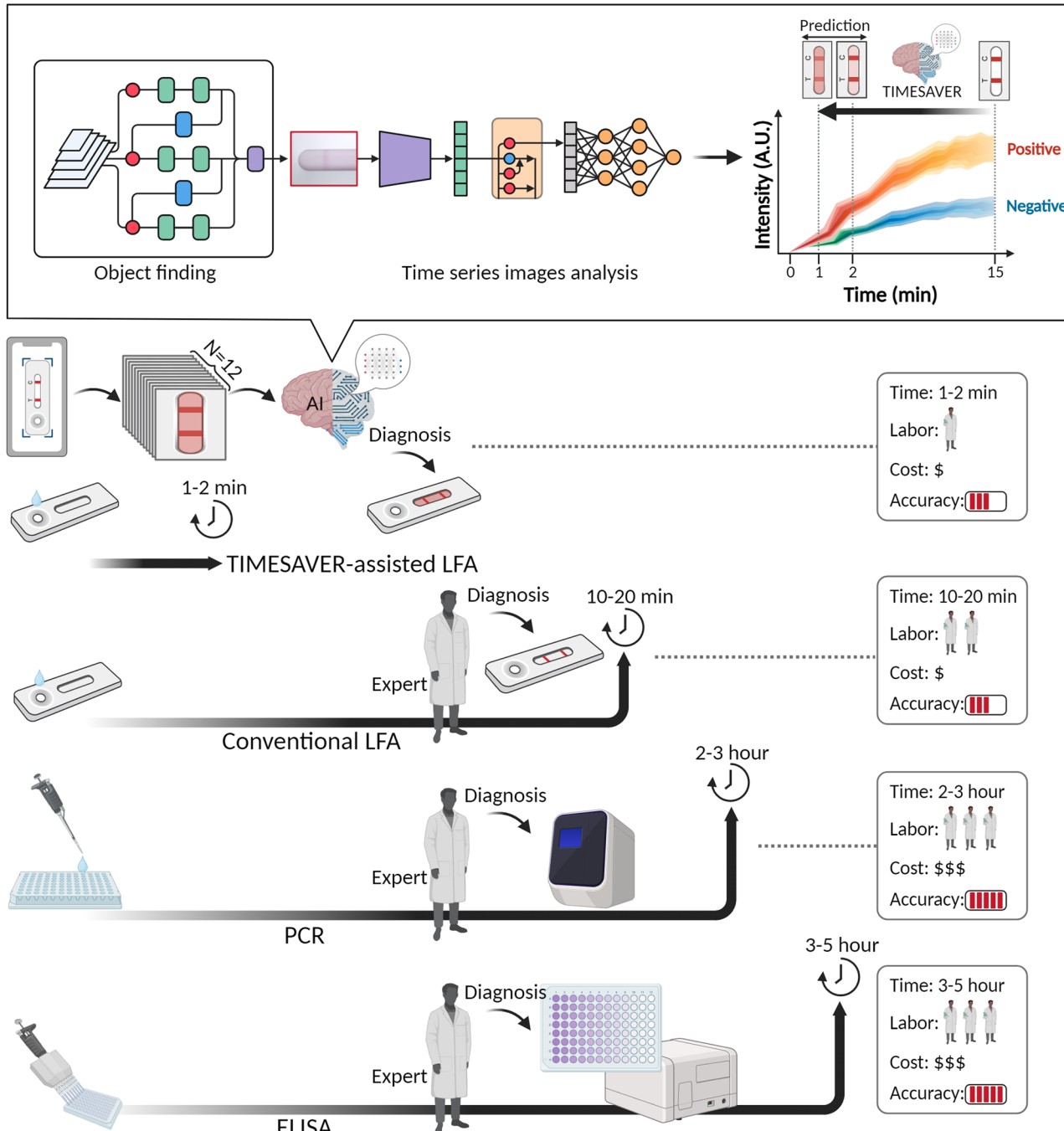

**Fig. 1 | Figure of merit of AI-powered TIMESAVER algorithm.** This schematic illustrates three representative commercialized diagnostic tools: commercial LFA, PCR, and ELISA, along with their performance metrics, including time, labor, cost, and accuracy. The TIMESAVER algorithm, utilizing a comprehensive time-series deep learning architecture, provides enhanced result analysis through AI-based verification, all within a rapid 1–2 min assay time, outperforming human experts with a 15-min assay. LFA, lateral flow assay; PCR, polymerase chain reaction; ELISA, enzyme-linked immunosorbent assay; TIMESAVER, Time-Efficient Immunoassay with Smart AI-based Verification; AI, artificial intelligence.

applications. Among the four frameworks evaluated (ResNet-18, ResNet-34, ResNet-50, DenseNet-121[39]), ResNet-50 exhibited the highest accuracy at 97.6%, surpassing the performance of shallow-layer models. In Fig. 2e, we optimized the LSTM model. When forecasting using time-series data, we employed advanced recurrent neural network (RNN) algorithms, including LSTM and gated recurrent unit (GRU). LSTM, a type of recurrent neural network, excels in handling sequential data and addresses the vanishing gradient problem by employing a sophisticated memory cell. LSTM achieved an accuracy of 97.6%, while GRU obtained 91.7%.

In Fig. 2f, we present the trade-off curve between root mean squared error (RMSE) and normalized graphics processing unit (GPU) memory consumption across various assay time frames, effectively illustrating the AI-based optimized assay time. Note that assay time refers to the sequential images used in training and testing. As we incorporated additional time-series data, the RMSE values were exponentially reduced, indicating enhanced accuracy. However, this improvement was accompanied by a linear increase in GPU memory consumption. Controlling GPU memory consumption is a key parameter for achieving optimal deep learning operation, as higher GPU

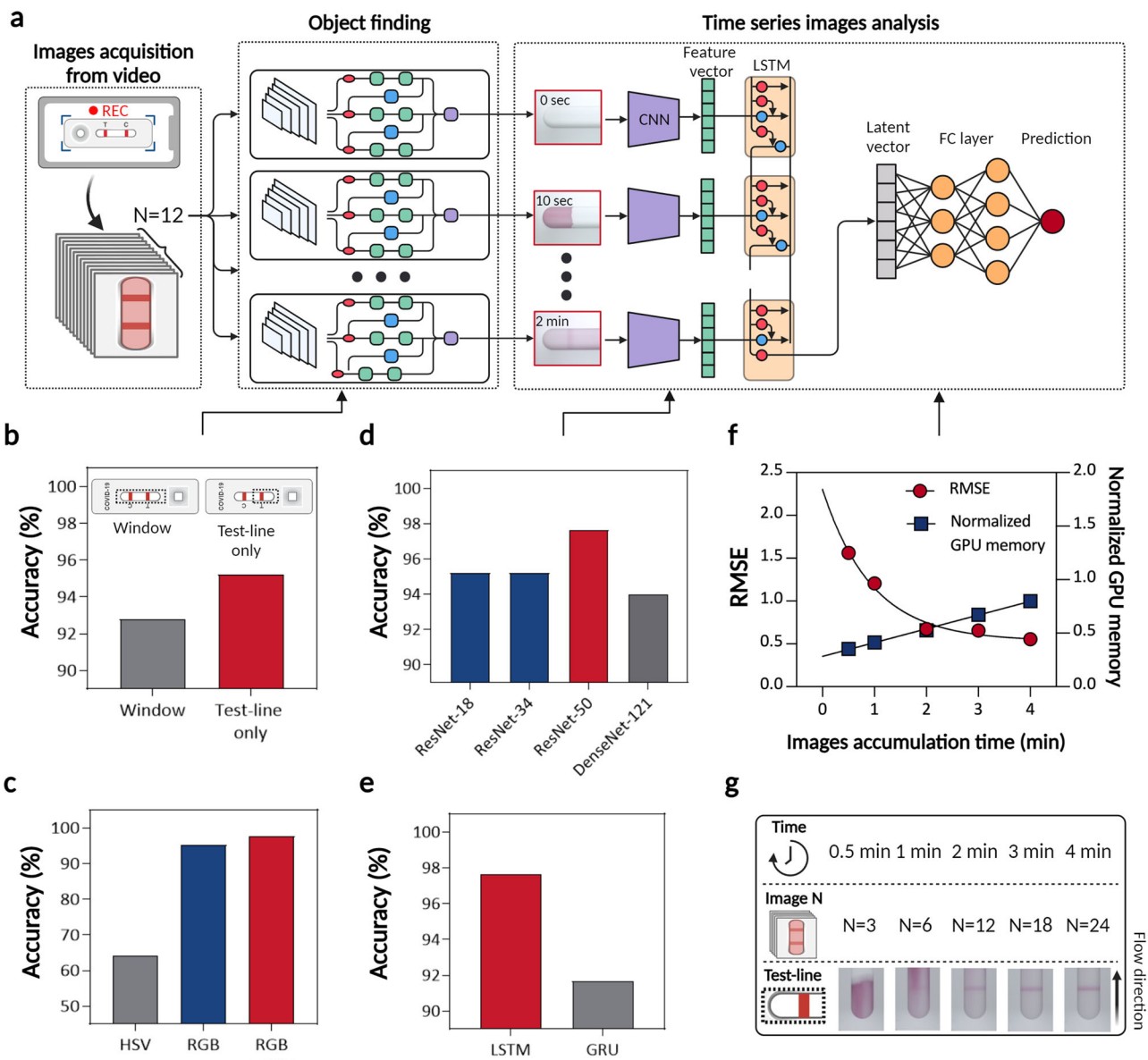

**Fig. 2 | Algorithm optimization. a** TIMESAVER algorithm comprises object detection using YOLO, time series image analysis through CNN-LSTM, and the FC layer. **b** Object Finding: Two ROI selection methods in LFA compared. Selecting only the test line resulted in higher accuracy (95.2%) compared to using the window (92.9%). **c** Data Augmentation using RGB and HSV: combining both yielded an accuracy of 97.6%. **d** CNN Model Optimization: ResNet-50 demonstrated the highest accuracy at 97.6%, outperforming other models. **e** LSTM optimization: LSTM achieved an accuracy of 97.6%, while GRU obtained 91.7%. **f** Trade-off between Root Mean Squared Error (RMSE) and normalized GPU memory consumption, illustrating the AI-based optimized assay time. **g** Time-series Images: Sequential images show the progression of the assay over time. TIMESAVER, Time-Efficient Immunoassay with Smart AI-based Verification; CNN, convolution neural network; LSTM, long short term memory; FC, fully connected; ROI, region of interest; GRU, gated recurrent unit; RMSE, root mean squared error; GPU, Graphics Processing Unit; AI, artificial intelligence.

memory consumption leads to longer training/test times and requires more expensive hardware. Consequently, we postulate that a 2-min time series may represent the optimal condition when employing the TIMESAVER model.

In Fig. 2g, we show the acquired images over time. After approximately 30 s, the samples loaded in the sample reservoir reached the test line, and the test line appeared after 1–2 min, depending on the concentration/titers of the target. All the images were taken at 10-s intervals, resulting 6 images acquired per minute. For example, in a 2-min assay, we trained on 12 sequential images, then tested sequential images with a 2-min assay time. Interestingly, in the time scale of 1 to 2 min, we observed unclear background signals with

the naked eye; however, the TIMESAVER model could detect the colorimetric signal with higher accuracy.

## Assay of infectious diseases via TIMESAVER

Figure 3 presents the assessment of infectious diseases, specifically COVID-19 antigen and Influenza A/B, using a 2-min assay facilitated by the TIMESAVER model. To assess the diagnostic accuracy of COVID-19 in Fig. 3a, we employed standard data (target protein spiked rapid kit running buffer) and trained the TIMESAVER model using our training set, which included both the training data ($n = 594$) and a validation subset (10% of the training set). We developed a regression model for TIMESAVER and categorized the regression values into five classes:

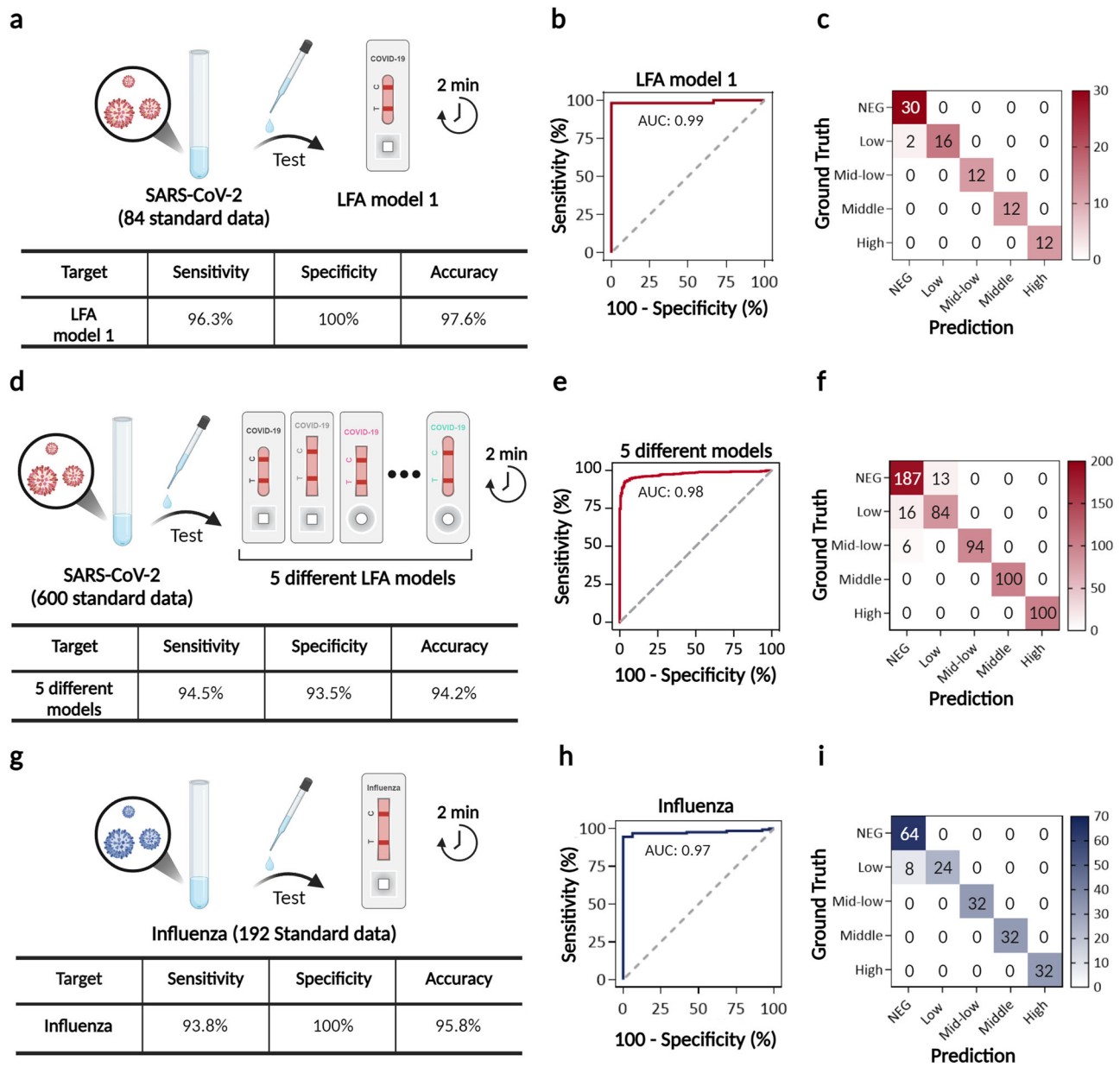

**Fig. 3 | Evaluation of a commercial LFA for infectious disease (COVID-19, Influenza A/B) with a 2-min assay using the TIMESAVER Model. a–c** COVID-19 assay: **a** TIMESAVER achieved a sensitivity of 96.3%, specificity of 100%, and accuracy of 97.6% from 84 data, demonstrating the model's proficiency in making initial decisions. **b** The ROC curve illustrates an AUC of 0.99. **c** The confusion matrix highlights that the false negatives were primarily associated with low-concentration samples. **d–f** Universality of COVID-19 assay by assessing its performance on various commercialized LFA Kits: **d** The average sensitivity, specificity, and accuracy across these five different models (n = 600), each with distinct form factors, were 94.5%, 93.5% and 94.2%, respectively. **e** The AUC value reached 0.98, as shown in the ROC curve. **f** The confusion matrix demonstrates that the ability to discriminate lower concentrations and negative controls is pivotal in LFA assays for achieving higher accuracy. **g–i** Influenza assay: **g** The sensitivity, specificity, and accuracy of the influenza model were 93.8%, 100%, and 95.8%, respectively. **h** The AUC value attained 0.97, as indicated by the ROC curve. **i** The confusion matrix indicates that the false negatives were predominantly linked to samples with low concentrations. TIMESAVER, Time-Efficient Immunoassay with Smart AI-based Verification; ROC, receiver operating character; AUC, area under curve; LFA, lateral flow assay.

high, middle, mid-low, low, and negative control. It's important to note that we categorized the images from data into these classes following the manufacturer's supplied guidelines, which are as follows: high (levels 8–7), middle (levels 6–5), mid-low (levels 4–3), low (levels 2–1), and negative control (level 0). The manufacturer's color chart is presented in Fig. S1. Utilizing standard data enables us to categorize into five classes, as opposed to the binary classification employed with clinical samples. Consequently, we can conduct a more comprehensive examination of the underlying causes of false positive and false negative signals. Since each dataset comprises 12 time frame images with 10-s intervals, the total number of images used for training was

7,128. We conducted tests with 84 data (54 positive and 30 negative). Our results indicate that the AI-based decision-making process, performed within 2 min, achieved a sensitivity of 96.3%, specificity of 100%, and accuracy of 97.6%, showcasing the excellence of the TIME-SAVER model in making initial decisions.

In Fig. 3b, c, we present receiver operating characteristic (ROC) curves and a confusion matrix for the 2-min assay of COVID-19 using the TIMESAVER algorithm. ROC curves provide a comprehensive view of the model's performance, with a higher area under the curve (AUC) indicating better classification ability. Our analysis revealed that the TIMESAVER model achieved an AUC of 0.99, affirming its excellent

performance as an assay classifier. The confusion matrix (Fig. 3c) highlights the critical nature of accurately diagnosing low-concentration data, a challenge even for experts when relying solely on visual inspection. We observed that the false negatives ($n = 2$) were caused by the low-concentration samples. This provides valuable insights for improving sensitivity and specificity. One viable strategy involves augmenting the training data. By incorporating more data with low concentrations, we can fine-tune sensitivity and specificity, as demonstrated in our previous paper[39]. In the following section, we will demonstrate the enhanced accuracy of our clinical assay. This will be achieved by integrating clinical data from 84 patients, including 13 with Ct values > 29, corresponding to low concentration/titer, and 32 healthy controls, as detailed in Fig. 5.

Universality is a key characteristic of the TIMESAVER algorithm. We validated its universality by assessing its performance on various commercialized LFA models (Fig. 3d–f). In this study, we tested an additional five LFA models ($n = 600$, Fig. 3d, Fig. S2, Supplementary Table 3). We exclusively trained the TIMESAVER model with an additional set of time-series data ($n = 300$) combined with the pre-existing dataset ($n = 594$), resulting in a total training dataset of 894. Given that each dataset consists of 12 time frame images, the total number of images used for training amounted to 10,728. To test the algorithm, we applied the TIMESAVER model initially trained with LFA model 1 (COVID-19 Ag LFA kits, Calth Inc.). Interestingly, the average sensitivity and specificity across these five different models ($n = 600$), each with distinct form factors, were 94.5% and 93.5%, respectively. The variation in performance can be attributed to differences in membrane types, designs, materials, flow rates, and other factors among LFA kits from various manufacturers. Such variations are expected due to the hardware-related disparities between these different LFAs. The AUC value reached 0.98 as shown in the ROC curve (Fig. 3e). Furthermore, from the confusion matrix (Fig. 3f), it is evident that the ability to discriminate lower concentrations and negative controls plays a pivotal role in LFA assays for achieving higher accuracy. We anticipate that further training with various LFA models will lead to increased accuracy, as demonstrated in our previous works[39].

We broadened our validation efforts to include influenza testing. The influenza kit in our study had A, B, and control lines, but due to limited sample availability, we only tested for influenza A. Illustrated in Fig. 3g, h, the manuscript details the sensitivity, specificity, and accuracy in detecting Influenza A, based on a dataset of 192 test samples. The influenza test kits exhibited a sensitivity of 93.8%, specificity of 100%, and an accuracy of 95.8%. The AUC value derived from the ROC curve was 0.97. It was observed that the false negatives ($n = 8$) were predominantly due to samples with low concentrations, which adversely affected sensitivity. However, the Lateral Flow Assay (LFA) enhanced by the TIMESAVER model demonstrated that it is possible to achieve a quick assay time while still maintaining the essential sensitivity and specificity for effective point-of-care diagnosis.

### Assay of non-infectious biomarkers for emergency room (ER) via TIMESAVER

Next, we further validated the performance of the TIMESAVER assay for non-infectious biomarkers, including Troponin I and hCG for ER. Initially focusing on Troponin I, as shown in Fig. 4a–c, we acknowledged its clinical relevance above 0.4 ng/ml, following previous research[17]. Therefore, we set a cut-off at 0.5 ng/mL and established a five-class multi-classification using recombinant protein, based on LFA manufacturer's guideline. This involved training with 618 data, validation with 62 data, and testing with 96 data. The results yielded a sensitivity of 96.9%, specificity of 98.4%, and accuracy of 97.9% (Fig. 4a). In Fig. 4b, the AUC value from the ROC curve was 0.99, and the TIMESAVER demonstrated high accuracy within a 2-min diagnostic timeframe. TIMESAVER showed some false signals at lower concentrations (Fig. 4c), which appear to be more a limitation of the LFA

rather than the algorithm. These results confirm the effectiveness of our algorithm in achieving multi-classification within just 2 min of testing, underscoring its utility in rapid diagnostic scenarios.

In emergency room settings, rapid diagnosis of hCG is essential, particularly for assessing pregnancy in patients. (Fig. 4d) demonstrates the sensitivity, specificity, and accuracy for hCG detection within 2 min, using test data ($n = 60$). The results revealed that the sensitivity, specificity, and accuracy for hCG were 97.5%, 95.0%, and 96.7%, respectively. The AUC value derived from the ROC curve was 0.95 (Fig. 4e), and the confusion matrix (Fig. 4f) suggests the effective performance of the classifier, even when applied in a 2-min assay utilizing the TIMESAVER model.

We aimed to assess the feasibility of achieving the assay within 1 min using commercially available diagnostic tests (Fig. 4g–i). Generally, hCG self-tests exhibit rapid flow velocity, and signal readings are typically recommended after a 5-min wait according to the manufacturer's guidelines. In our primary training data ($n = 594$), initially trained for COVID-19, we incorporated an additional hCG dataset ($n = 24$), resulting in a total training set of 618 data (Fig. 4g). The hCG dataset consisted of 30 images captured at 2-s intervals. We then used 12 images taken between 36 and 60 s. The test dataset consisted of 94 standard data. Even with a 1-min assay facilitated by TIMESAVER, we achieved a sensitivity of 90.6%, specificity of 93.3%, and an overall accuracy of 91.5%. The sensitivity, specificity, and overall accuracy of five human experts at 5 min were 90.9%, 87.3%, and 89.8%, respectively. In Fig. 4h, we observed that the accuracy with TIMESAVER at 1 min surpassed the accuracy of five experts at 5 min. As anticipated, false positives and false negatives of TIMESAVER at 1 min were primarily associated with lower concentrations (15 mIU), particularly those near the cutoff threshold (Fig. 4i).

### Blind tests using clinical samples

Figure 5 illustrates the clinical evaluation of COVID-19 through blind tests. We assessed the blind tests from three different groups: untrained individuals, human experts, and TIMESAVER, utilizing 252 test data (156 positives and 96 negatives). Clinical samples were collected from COVID-19 patients at Seoul St. Mary's Hospital, including SARS-CoV-2 patients ($n = 52$) and healthy controls ($n = 32$). The 252 test data come from the three different rapid kit tests performed on COVID-19 patients ($n = 84$). This information encompassed sample collection details, variants, sex, ages, and Ct values (Supplementary Tables 4, 5). All samples underwent RT-qPCR analysis, followed by the LFA assay. The data from the LFA assay were classified into five groups: high/middle/middle-low/low titer, and negative control, using a color chart level (high with levels 8–7, middle with levels 6–5, middle-low with levels 4–3, and low titer with levels 2–1 for positive, and negative with level 0). Among the positive data ($n = 156$), we distributed the data across four groups (high: 30, middle: 48, mid-low: 39, low: 39). We also included negative data from healthy controls ($n = 96$).

For the blind test, ten untrained individuals and ten human experts each tested 252 data, including 30 high, 48 middle, 39 middle-low, 39 low, and 96 negative data. This resulted in a total of 5040 blind tests for both untrained individuals and human experts. As shown in Fig. 5a, the colorimetric assay results were captured using a custom-made charge-coupled device (CCD) camera, or potentially a smartphone camera, displaying clear positive images in high and middle concentrations. However, below the mid-low concentration, no distinct positive signal could be captured. Interestingly, the assay conducted within 2 min exhibited a larger background signal, which hindered the clear observation of the colorimetric signal by the naked eye.

We presented the results of blind tests using images from a 15-min assay (Fig. 5b) followed by a 2-min assay (Fig. 5c), involving both untrained individuals and human experts, as well as the TIMESAVER algorithm, which demonstrated a notable reduction in assay time. The

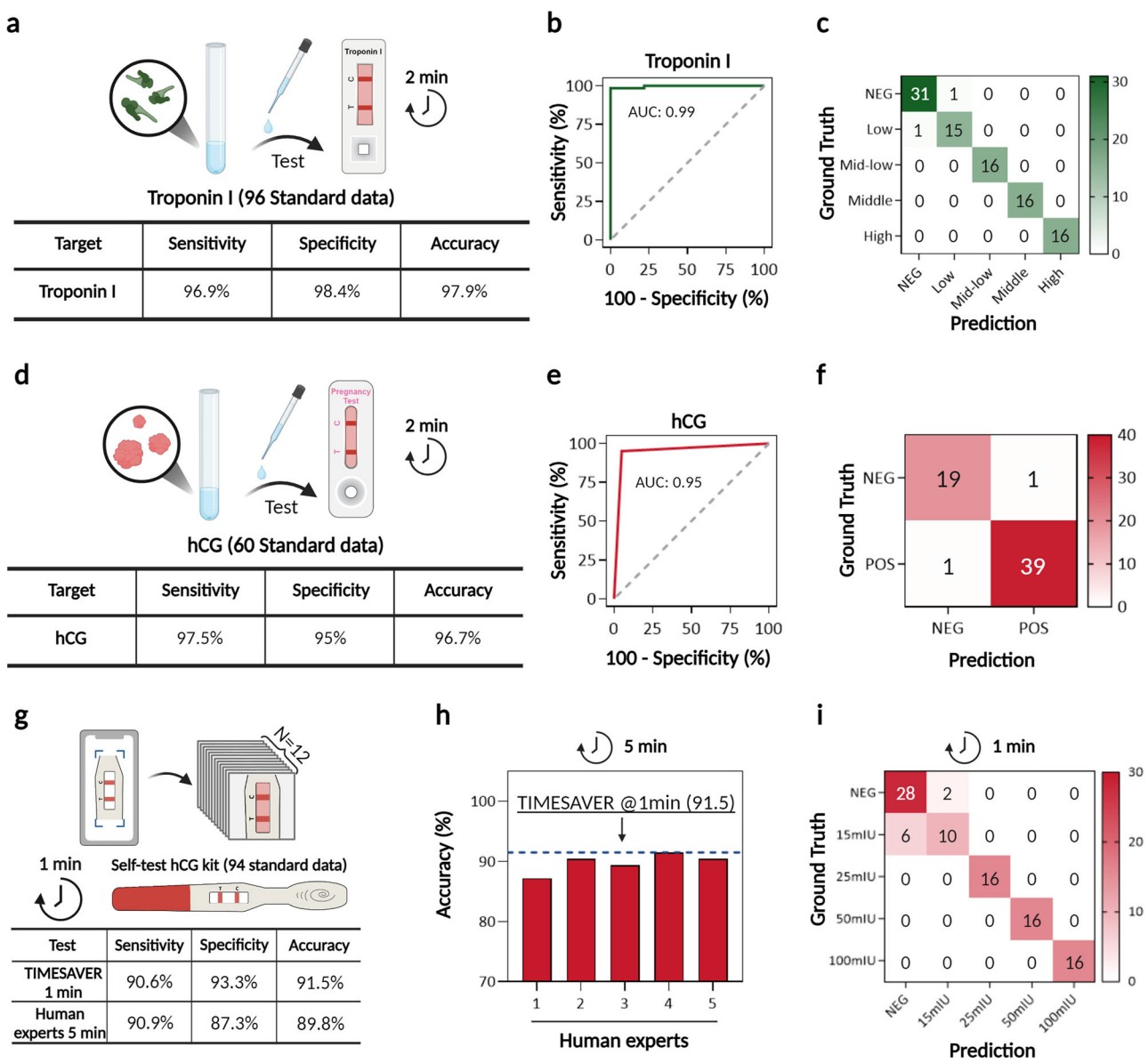

**Fig. 4 | Evaluation of non-infectious biomarkers (Troponin I and hCG) using the TIMESAVER Model. a–c** Troponin I assay: **a** The sensitivity, specificity, and accuracy for the detection of Troponin I were 96.9%, 98.4%, and 97.9%, respectively. **b** The AUC from the ROC curve was 0.99. **c** The confusion matrix between ground truth (y-axis) and predicted label (x-axis). **d–f** hCG test: **d** hCG detection achieved sensitivities of 97.5%, specificities of 95%, and an accuracy of 96.7% from 60 test data. **e** The ROC curve produced an AUC of 0.95. **f.** The confusion matrix. **g–i** Evaluating the feasibility of a 1-min assay with hCG self-tests from 94 test data:

**g** With a 1-min assay with TIMESAVER sensitivities of 90.6%, specificities of 93.3%, and an overall accuracy of 91.5% were achieved. **h** The accuracy with TIMESAVER at 1 min surpassed the accuracy of five experts measuring at 5 min. **i** The confusion matrix illustrates that false positives and negatives were predominantly associated with lower concentrations. TIMESAVER, Time-Efficient Immunoassay with Smart AI-based Verification; hCG, human chorionic gonadotropin; ROC, receiver operating character; AUC, area under curve; LFA, lateral flow assay.

15-min assay shown in Fig. 5b was conducted following the manufacturer's guidelines for conventional assays. In these 15-min assay images, untrained individuals reached an accuracy rate of 70.7%, while human experts attained 78.1%. The lower accuracy compared to the manufacturer's claim of >90% sensitivity and >99% specificity can be attributed to our inclusion of a substantial number of lower titer data. Nevertheless, the TIMESAVER model surpassed both human experts and untrained individuals in performance, achieving a higher accuracy of 80.6% even in a shortened 2-min assay.

When the assay time was reduced to 2 min (Fig. 5c), identifying clear positive signals for mid-low concentrations became problematic for the naked eye, and the reddish background often led to more false positives. As a result, the accuracy rates for untrained individuals and human experts fell to 59.4% and 64.6%, respectively. In contrast, the

TIMESAVER algorithm maintained a high accuracy of 80.6% in the 2-min assay. While the accuracy of human interpretation significantly decreased at lower concentrations (lower viral load), indicating a tendency for human error in rapid assessments, the AI-driven TIME-SAVER algorithm showed greater precision, effectively handling background noise and unclear colorimetric signals. This allowed for fast assays with improved accuracy, showcasing the potential of AI in enhancing rapid diagnostic techniques.

Figure 5d displays the influence of clinical training data on ROC curves. Initially, we present ROC curves trained with a standard dataset (n = 594, shown in blue and labeled as 'standard only'). We then demonstrate improved ROC curves achieved after additional training with clinical data (n = 694, shown in red and labeled as 'standard and clinical'). The ROC curve is a widely used tool for assessing the clinical

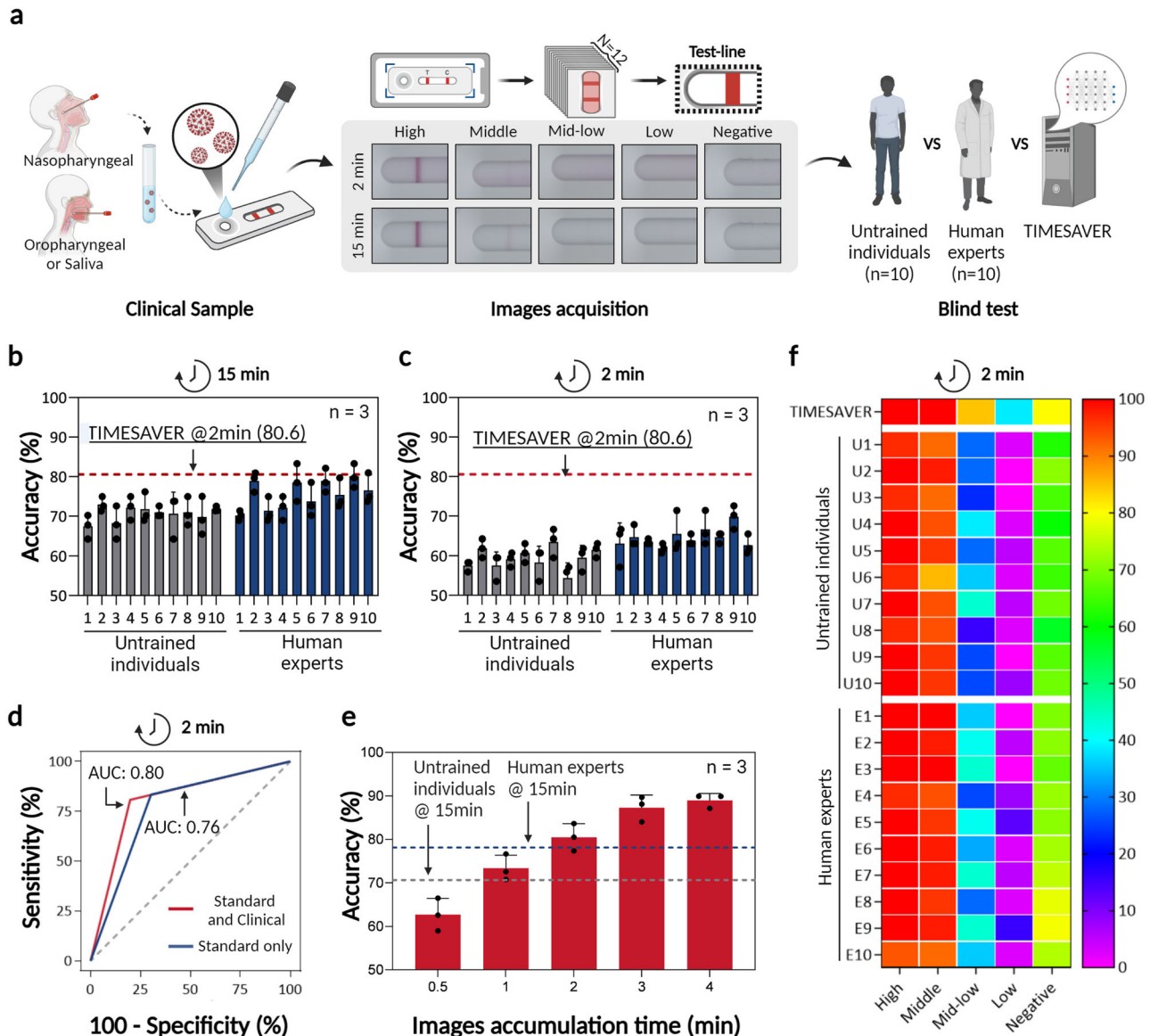

**Fig. 5 | Clinical validation via blind tests. a** In the blind test, ten untrained individuals and ten human experts assessed 252 data, including various concentration levels and negative data from COVID-19 clinical samples. **b, c** Blind test results from a 15-min assay and a 2-min assay demonstrate significant fast assay with TIMESAVER ($n = 3$): **b** TIMESAVER with 2-min assay achieved higher accuracy (80.6%) compared to human experts (78.1%) and untrained individuals (70.7%) in the 15-min assay. **c** In the 2-min assay, TIMESAVER maintained high accuracy (up to 80.6%), while untrained individuals and human experts experienced lower accuracy rates (59.4% and 64.6% respectively). **d** Additional clinical data improved AUC (0.80) compared to the standard dataset alone (0.76). **e** With a 2-min assay, TIMESAVER outperformed human experts with 15-min assay, achieving an accuracy rate of 80.6% (compared to 78.1%) ($n = 3$). **f** The heat map shows that in the mid-low titer range, TIMESAVER demonstrated an accuracy rate of 84.6%, surpassing both untrained individuals (29.2%) and human experts (37.2%). Error bars represent standard deviation from the mean. TIMESAVER, Time-Efficient Immunoassay with Smart AI-based Verification; hCG, human chorionic gonadotropin; ROC, receiver operating character; AUC, area under curve; LFA, lateral flow assay.

effectiveness of diagnostic models. The AUC with the inclusion of clinical data (0.80) exceeded that with the standard dataset alone (0.76). Although the TIMESAVER algorithm with a 2-min assay might not entirely match the accuracy standards of clinical laboratories, its ability to continuously improve diagnostic accuracy through learning from acquired images is notable. By further incorporating deep learning with clinical samples, we can enhance the clinical accuracy of our diagnostic approach.

We demonstrate the capability of TIMESAVER to achieve accuracy levels comparable to those of human experts in the shortest possible time frame (Fig. 5e). We initiated the assay timer when the sample was introduced into the sample reservoir, capturing sequential images over time. We established five distinct datasets, each representing varying assay durations (0.5, 1, 2, 3, and 4 min). For example, in the case

of a 1-min assay, we obtained 6 images with 10-s intervals. Our TIMESAVER model demonstrated that it requires only 1 min to attain accuracy equivalent to that achieved by untrained individuals. With a 2-min assay, we achieved an accuracy rate of 80.6%, surpassing the accuracy of human experts at the 15-min mark (78.1%). As depicted in Fig. S3, the samples reached the test line within 1 min, enabling the AI to precisely ascertain the assay results during the initial color development phase. In comparison to conventional human-conducted assays, where TIMESAVER completes the assay in just 2 min, it consistently outperforms human experts in terms of accuracy.

The heat map indicates that human visual assessment, conducted by both untrained individuals and experts, shows a decrease in accuracy, particularly within the mid-low titer ranges (Fig. 5f). In the mid-low titer category, untrained individuals managed an average accuracy

of only 29.2%, while human experts fared slightly better at 37.2%. In contrast, our algorithm achieved an accuracy rate of 84.6%. For the low titer category, the accuracy was even lower, with untrained individuals at 2.8% and human experts at 5.4%, but our deep learning algorithm significantly outperformed at 38.5% accuracy. In cases of high and middle titer concentrations, the TIMESAVER algorithm consistently provided reliable and accurate data, effectively eliminating the variability seen in human visual assessments.

## Discussion

In diagnostics, the twin goals of achieving high sensitivity and rapid processing are crucial for POCT. The TIMESAVER system, with its three core components—object detection, CNN-LSTM networks, and FC layers—exemplifies its proficiency in delivering results within a notably short span of 2 min. This achievement underscores a range of significant practical benefits associated with the TIMESAVER algorithm:

1. Fast assay: We have introduced one of the fastest assays among commercially available biofluid-based tests (Supplementary Table 1-2). Our system is capable of delivering results in just 1–2 min, rivaling the accuracy achieved by human experts in a traditional 15-min assay. This feature is made possible by the use of the TIMESAVER algorithm, which learns from the initial colorimetric image changes and makes decisive determinations in these crucial early stages.

2. Universality Across Diverse LFA Models and Targets: The validation of the TIMESAVER algorithm across various LFA models for both infectious and non-infectious biomarkers highlights its broad applicability. Capitalizing on the sequential image methodology, we envision extending this adaptable algorithm to smartphones. Given the global population of over 6 billion smartphone users, the fusion of TIMESAVER with smartphone technology has the potential to revolutionize point-of-care testing (POCT), making it more accessible and affordable.

3. Time-Domain Decision Framework: At the core of our approach is an algorithm based on time-domain decisions, specifically designed for assays dependent on binding kinetics. This encompasses techniques such as surface plasmon resonance (SPR), isothermal titration calorimetry (ITC), and fluorescence resonance energy transfer (FRET). Our algorithm's versatility enables the potential adaptation to a variety of sensing technologies, including nanowire sensors, field-effect transistor (FET) sensors, and digital immunoassays, broadening its utility across diverse biomedical research and clinical environments. Additionally, we are advancing the integration of this framework with sophisticated biosensing methods like electrochemical impedance spectroscopy (EIS) and quartz crystal microbalance (QCM), with the aim of offering comprehensive analytical capabilities across a wide range of biomolecular interactions.

4. Implications for Public Health: Strategies for reducing assay times hold transformative potential, not only in enhancing individual patient care and treatment but also in bolstering public health. We have demonstrated the versatility of LFAs in early detection, covering a spectrum from infectious diseases to non-infectious biomarkers. This achievement has the potential to significantly impact diagnostics, by providing healthcare professionals with the agility to make rapid decisions.

In the continually evolving field of medical diagnostics, the TIMESAVER algorithm represents a significant advancement in the realm of rapid and reliable assays. The convergence of speed, accuracy, and affordability presents a promising path for the future of healthcare through AI-based POCT.

## Methods

### Ethical statements

Samples of individuals diagnosed with COVID-19 were collected at Seoul St. Mary's Hospital from April 2021 to May 2022. Approval for this study was granted by the institutional review board

(KC21TIDI0134K) at Seoul St. Mary's Hospital, and participants provided informed consent to be involved in the research study, including the sharing of individual-level and potentially identifying data.

### Infectious disease: COVID-19 antigen

This study used commercially available Lateral Flow Assay (LFA) kits for COVID-19 antigen detection (Calth Inc., Republic of Korea), strictly following the manufacturer's protocol. We introduced a 100 μL sample into each LFA kit and captured images at 10-s intervals over various time periods (0.5, 1, 2, 3, and 4 min). For detailed data analysis, we developed a custom LFA reader using LabVIEW v2019 SP1 (National Instruments Co., USA). A blind test was conducted to compare the TIMESAVER-AI's performance with human analysis, involving both trained experts and untrained individuals. They made diagnoses based on data from the LFA reader, typically at a fixed time point of around 15 min.

For the COVID-19 assay, we used the SARS-CoV-2 virus Nucleocapsid protein (FPZ0516, Fapon Biotech) as a standard. Eight different concentrations were prepared by diluting the protein in the kit's running buffer at 1/2-fold increments, from 50 ng/mL to 0.39 ng/mL. We obtained positive lateral flow assay data, categorized into four colorimetric classes (High, Middle, Mid-low, and Low) using these serially diluted samples. Negative control data were obtained solely from the kit's running buffer. The COVID-19 Ag LFA kits were utilized to generate both positive and negative data from all standard samples. Classification into five classes (High, Middle, Mid-low, Low, Negative) was based on the color chart provided by the manufacturers, commonly used in LFA production and evaluation. The collected data included both positive and negative samples, captured as a time series at 10-s intervals.

### Infectious disease: COVID-19 Ag and its universality

To evaluate the versatility of our deep learning model for COVID-19, we assembled a dataset using data from five distinct LFA models. The COVID-19 Ag kits incorporated in this study were obtained from five different manufacturers: Panbio COVID-19 Ag (Abbott, USA), GENEDIA COVID-19 (GCMS, Republic of Korea), COVID-19 Ag Test (Humasis, Republic of Korea), COVID-19 Ag (Genbody, Republic of Korea), and InstaView COVID-19 (SGmedical, Republic of Korea). For generating time-series data related to COVID-19, we carefully diluted the nucleocapsid protein (N-protein) target into eight concentrations using the running buffers provided in the kits from each manufacturer. Since the limit of detection (LOD) for N-protein varies among manufacturers, we adjusted the eight sample concentrations based on each kit's specified LOD. Our analysis involved categorizing the test line's colorimetric readings into four classes (High, Middle, Mid-low, Low). Negative data were derived solely using the running buffer from each kit. Both positive and negative data sets were methodically captured as time-series at 10-s intervals. This diverse dataset allowed us to evaluate the adaptability and efficacy of our deep learning model across various COVID-19 Ag kits.

### Infectious disease: Influenza A/B

For the influenza A/B tests, we employed commercialized kits: the influenza A/B test kit (Daewoong Pharmaceutical Co., Republic of Korea) as LFA kits. We utilized the influenza virus nucleocapsid protein (RDR-187, Medix Biochemica) as the standard sample. We prepared eight different concentrations by diluting the sample in the kit's running buffer at 1/2-fold increments, ranging from 10 ng/mL to 0.3 ng/mL. Positive data for the lateral flow assay was obtained using serially diluted samples with varying concentrations prepared in the running buffer. Negative control data was exclusively acquired using the running buffer from the kit. The influenza LFA kits were employed to extract both positive and negative data from the standard samples.

## Non-infectious biomarkers: Troponin I

For the Troponin I tests, we utilized commercialized kits (Humasis, Republic of Korea). CNTI recombinant antigen (GRNCTNIN101, Fappon) was employed as a control material and serially diluted using human serum (H3667, Sigma Aldrich). We prepared samples covering a broad concentration range, ranging from 0.5 ng/ml to 20 ng/ml (clinically relevant range: >0.4 ng/ml). Negative data were exclusively derived using the running buffer from each kit. Both positive and negative datasets were systematically captured as time-series at 10-s intervals.

## Non-infectious biomarkers: hCG

For the hCG tests, we utilized commercially available AllCheck hCG card kits (Calth Inc., Republic of Korea) as LFA kits. Early detection of hCG, particularly in the first trimester of pregnancy, is crucial, especially in emergency situations where pregnancy might not be immediately evident. Therefore, our aim was to determine the effectiveness of the hCG test in identifying early-stage pregnancy.

In our hCG experiments, we used hCG standard control material (Calth Inc., Republic of Korea), available in three different concentrations: high (250 mIU/mL), low (25 mIU/mL), and negative (0 mIU/mL). To cover a broad concentration spectrum, we prepared eight distinct hCG levels through serial dilution. This included three concentrations representing early pregnancy (within ~3 weeks) ranging from 0.375 mIU/mL to 15 mIU/mL, and three additional concentrations for the subsequent early stages of pregnancy (6-8 weeks) from 50 mIU/mL to 200 mIU/mL. Overall, we obtained data for eight different concentrations from the positive hCG samples. Both positive and negative data were gathered using the AllCheck hCG kit. For hCG tests, positive and negative data sets were collected as time series at 10-s intervals over 15 min.

For the 1-min hCG experiments, we utilized hCG standard control material (international standard 5th, NIBSC 07-364). To obtain comprehensive data covering various concentrations, we prepared 1-min hCG samples at four different levels using 1X tris-buffered saline (TBS) (1% bovine serum albumin) running buffer. The positive range of 1-min hCG samples prepared at four different concentrations is 100 mIU, 50 mIU, 25 mIU, and 15 mIU, while negative data were generated by introducing only running buffer. All positive and negative 1-min hCG data were obtained using the Signal Q Pregnancy Test Kit (Daewoong Pharmaceutical Co., Republic of Korea). This diagnostic LFA device provides results in 5 min, allowing for a relatively fast assay speed. Therefore, we collected time-series data at 2-s intervals for 1 min.

## Clinical data of COVID-19 blind test

For additional training data, we formed a dataset using clinical data. To acquire COVID-19 clinical data, SARS-CoV-2 virus clinical samples were collected from two different specimens: NP/OP (Nasopharyngeal and Oropharyngeal) swabs and saliva, using COVID-19 Ag LFA kits (Calth Inc., Republic of Korea). Positive clinical samples were obtained from NP/OP swabs of 40 patients and saliva samples of 12 patients, each with varying titers. Conversely, negative clinical samples were gathered from 20 individuals using NP/OP swabs and from 12 individuals using saliva samples as part of the healthy control group. All positive clinical samples were classified into four classes (High/Middle/Mid-low/Low) based on the colorimetric chart criteria of the LFA test line. Both positive and negative data were collected as time-series data at 10-s intervals over a 15-min duration. This clinical dataset was then incorporated into the training process to enhance the performance and accuracy of the TIMESAVER algorithm.

## Blind test of TIMESAVER AI and humans

In the COVID-19 Ag blind test, we aimed to compare the predictive performance of the TIMESAVER AI model with that of both 10 untrained individuals and 10 experts skilled in interpreting LFA results,

utilizing clinical data. Based on the aforementioned clinical data, we conducted a blind test involving 40 NP/OP (Nasopharyngeal and Oropharyngeal) swab samples, which were subsequently categorized into four classes: High titer ($n = 8$), Middle titer ($n = 12$), Mid-low titer ($n = 11$), and Low titer ($n = 9$). Additionally, 12 saliva samples were classified as follows: High titer ($n = 2$), Middle titer ($n = 4$), Mid-low titer ($n = 2$), and Low titer ($n = 4$). Using COVID-19 Ag LFA kits from Calth Inc., Republic of Korea, we conducted ($n = 3$) assays for each clinical sample, resulting in positive samples being classified into four classes, while negative samples provided time-series data from the test lines. Subsequently, the test line images at the 15-min mark were compiled into a series of blind test questions and presented to both the 10 untrained individuals and the 10 human experts. In total, the 20 participants underwent 180 blind tests each for the NP/OP samples and an additional 72 blind tests for the saliva samples.

## Data preparation for deep learning models: training, validation, and testing

We organized three distinct datasets: training, validation, and testing, crucial for optimizing and assessing the deep learning model's performance. The training dataset was employed to refine the model, the validation dataset served to prevent overfitting, and the test dataset was instrumental in the final evaluation. Before training began, a random 10% of the training dataset was allocated as the validation dataset to ensure an unbiased assessment.

In order to effectively show TIMESAVER's capacity to expedite the assay process, it was imperative to capture the temporal evolution of images. Consequently, we prepared a training set of 694 data and a test set of 252 data, each encompassing 12 image frames for COVID-19 test. For the testing phase, we utilized a set of clinical samples, consisting of 52 positive and 32 negative samples. For the universality test of different LFA models, we collected additional time-series datasets from the COVID-19 Ag LFA tests of the five manufacturers, resulting in 894 training data and 600 test data for each manufacturer. In the Influenza A/B test, we employed 642 datasets for training and allocated 192 datasets for testing. The Troponin I test utilized 618 datasets for training and 96 for testing. For the hCG test (expert), 624 training datasets were used, accompanied by 60 datasets for the test.

We developed a 1-min assay, as illustrated in Fig. 4g–i, capturing images every 2 s for 1 min, and then selecting 12 representative images from the 36- to 60-s timeframe. All-time series data sets were collected under consistent lighting conditions. This controlled setting was crucial for accurately capturing essential information regarding fluid dynamics, distribution characteristics, and the temporal color variations in both test and control lines. The trivalent LFA assays were performed using a custom-designed reader, with data acquisition extending over a 5-min span at 2-s intervals. Following this, our specially devised algorithm was applied to isolate and analyze the test lines.

As discussed in our previous work[39], we have extensively addressed the challenges related to invalid test results and their decision-making processes. For this study, we used single images of the control line captured at either 1 or 2 min. The control line region was cropped to assist in determining the validity of the test strip.

## Deep learning models

The architecture of our model is structured into two stages: image feature extraction and time-series analysis. This bifurcation is necessary due to the use of different foundational models: Convolutional Neural Networks (CNN) for extracting salient features from images and Recurrent Neural Networks (RNN) for analyzing time-series data. Specifically, for image feature extraction, we utilize the renowned ResNet-50 architecture. This allows for the encoding of high-dimensional images into lower-dimensional features while retaining

their spatial attributes, which are crucial for the subsequent analytical stage.

To effectively train ResNet-50, we employed pretrained weights available in PyTorch, which were then fine-tuned to fit our unique dataset. These weights were originally trained on the large-scale IMA-GENET1k dataset, designed for classifying images into 1000 different categories. Leveraging a pre-trained model and fine-tuning it for our specific needs enables us to develop a robust feature extractor, even with a relatively smaller dataset at our disposal.

For the time-series analysis component, we crafted a Recurrent Neural Network (RNN) specifically for LFA kit diagnostics, based on the Long Short-Term Memory (LSTM) architecture. This model outputs real-number predictions for concentration estimation. In our model, positive and negative samples are numerically ranked in descending order according to their concentration, with higher concentrations assigned larger numbers.

To predict concentrations accurately, we implemented the Mean Absolute Error (MAE) loss function (Eq. 1). While both Mean Squared Error (MSE) and MAE are common choices for regression loss functions, we opted for MAE due to its robustness, particularly in managing outliers. This approach ensures that our model is not only effective in making accurate predictions but also resilient in handling diverse data variations.

For the task of classifying samples as positive or negative, our method employs a threshold to differentiate between the two categories. Samples surpassing the set threshold are identified as positive, and those below it are classified as negative. For instances where the data is missing concentration information or the target differs from existing datasets, necessitating additional learning, we employ extra methodologies. Initially, we train the model using existing data. Then, we load the pretrained weights and modify the final fully connected (FC) layer's output dimension from 1 to 2, initializing the weights of this last layer. After this modification, we fine-tune the model using the Cross Entropy (CE) loss function (Eq. 2). This strategy significantly reduces learning time for non-standard data formats and enhances the model's versatility to handle a broader range of scenarios.

$$MAE = \frac{1}{N}\sum_{i}^{N}|y_i - \hat{y}_i| \qquad (1)$$

$$CE = -\sum_{i=1}^{2} y_i \cdot \log y_i \qquad (2)$$

**Statistics and reproducibility**

The error bars presented in the figures represent the mean ± SD. We repeated all the experiments at least thrice per point and analyzed the data using Microsoft Exel, Prism v 8.0. Biorender, Adobe Photoshop v 2020, and Adobe Illustrator v 2020 software were used for graphical analyses. No statistical method was used to predetermine sample size. No data were intentionally excluded from the analysis.

**Reporting summary**

Further information on research design is available in the Nature Portfolio Reporting Summary linked to this article.

## Data availability

Source data are provided with this paper. Example images used in this study are available at https://zenodo.org/records/10582232[46]. Source data are provided with this paper.

## Code availability

The overall source codes used in this study is available at: https://github.com/Artinto/Rapid_Deep_Learning-Assisted_Predictive_

Diagnostics_for_Point-of-Care_Testing which is archived in https://zenodo.org/records/10582339[47].

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

## Acknowledgements

This work was supported by the Bio & Medical Technology Development Program of the National Research Foundation funded by the Korean government (MSIT) (No. 2023M3E5E3080743). The present research has been conducted by the Excellent researcher support project of Kwangwoon University in 2023. S. Lee was funded by the Hyundai Motor Chung Mong-Koo Foundation and supported by Oneness Mission Club (OMC). We also thanks to Sunmok Kim, Jiwon Moon, and Hakjun Lee for his contribution for AI models. Illustrations in Figs. 1, 2, 3, 4, 5 were created with BioRender.com.

## Author contributions

S. Lee, J. S. Park, H. Woo, K. B. Lee, and J. H. Lee conceived and designed the study. S. Lee and J. S. Park acquired the images. H. Woo was responsible for the programming. D. Lee provided the kit and interpreted the results. S. Chung, D. S. Yoon, Y. K. Yoo, K. B. Lee and J. H. Lee wrote the manuscript. All the authors discussed the results and commented on the manuscript.

## Competing interests

The authors declare no competing interests.
