## [Peer Review File · Nature Communications]

Reviewers' comments:

Reviewer #1 (Remarks to the Author):

Title: Transforming Point-of-Care Diagnostics: 1-Minute Assays Enabled by Time-Series Deep Learning

Summary: The authors used well-trained time-series deep learning architecture and AI verification on the LFA platform, which enables 1-minute immunoassays of SARS-CoV-2 N-protein, influenza A/B N-protein, and hCG. While, there is serious homogeneity between the author's previously published papers (Nat Commun.,2023, 14, 2361) and the current manuscript, which compromises the originality of this work. Besides enhancing the performance of deep learning and AI, the authors should pay more attention to improving the analytical chemistry capabilities of AI-assisted LFA strips with new sensing elements. Hence, I cannot recommend the acceptance of this manuscript in this journal in present state.

Comment:

1. What is the innovation or advancement of this work in contrast to the authors' previous publications (Nat Commun. 2023, 14, 2361) except for two additional targets?
2. 10 pg/mL of SARS-CoV-2 N-protein, that is, the COVID-19 diagnostic cut-off value (Nat Commun. 2021, 12, 1931; J Clin Microbiol. 2021, 59(10), e01001-21), should be analyzed by the author's method.
3. Whether the time-series deep learning algorithm on the LFA platform has the potential to replace PCR.
4. Utilizing HCG as the detection target cannot highlight the necessity for a 1-minute assay. Authors may evaluate the performance of the AI-LFA for acute disease diagnostics, such as by choosing the biomarkers of myocardial infarction, acute renal failure, or toxic shock syndrome as the detection target.
5. Using "transforming" to promote the significance of deep learning-assisted LFA is overexaggerated. And, the AI-LFA cannot be called a revolutionary POCT. The aid of AI increases the learning cost for general people, whether an irregular photograph affects the accuracy of detection results, such as mobile phone manufacturer, exposure time, focus, and resolution.
6. To solidify the necessity of the AI-LFA strip, authors may develop a multiplexing LFA strip using fluorescent materials with different excitation wavelengths, since the naked eye-based assay has a limited ability to distinguish the gradients of various colors.
7. Generally speaking, the control line can confirm the strips work correctly and act as an Internal reference signal to improve the detection accuracy. ROI selection (Figure 2b) requires more data support and principal explanation than present results.
8. In the test strip, the control line confirms that the test paper is working correctly, while the test line shows whether the samples contain the target. This paper only obtains the diagnostic results through the test line. How do we exclude the wrong results caused by the failure of the test strips (invalid state)?
9. Very little data is obtained through a 1-minute analysis in this work. Is it appropriate to use the word 1-minute analysis in the title?

Reviewer #2 (Remarks to the Author):

The authors provide a deep learning methodology to generate an algorithm allowing the radical shortening of assay times for lateral flow assays, and present convincing AUC data for three assays demonstrating that the approach competes well with the conventional (end-point) observation for the same assays.

There is value in shorter assay times; furthermore, especially for home tests such as covid tests impatient users commonly, and without AI assistance, interpret assay results prematurely (essentially operating them outside of approved guidelines), with unclear consequences. A device allowing the interpretation much earlier would alleviate such misuse.

What is not shown, however, is a comparison of the results obtained with the time saver algorithm compared to say, visually reading the LFAs after a shortened time without the benefit of the timesaver algorithm or a phone imager. For many LFAs, especially covid antigen LFAs and higher viral loads, test lines appear very quickly, often within a minute or two, and results could be possibly conclusively interpreted just visually, with good AUC compared to running the same sample sets to normal conclusion. I wonder if the authors could comment on that and maybe present data on the specific value of the timesaver algorithm over just visually interpreting the LFAs after a shortened time.

Response letter

Reviewer's comments:

Reviewer #1 (Remarks to the Author):

Title: Transforming Point-of-Care Diagnostics: 1-Minute Assays Enabled by Time-Series Deep Learning

Summary: The authors used well-trained time-series deep learning architecture and AI verification on the LFA platform, which enables 1-minute immunoassays of SARS-CoV-2 N-protein, influenza A/B N-protein, and hCG. While, there is serious homogeneity between the author's previously published papers (Nat Commun.,2023, 14, 2361) and the current manuscript, which compromises the originality of this work. Besides enhancing the performance of deep learning and AI, the authors should pay more attention to improving the analytical chemistry capabilities of AI-assisted LFA strips with new sensing elements. Hence, I cannot recommend the acceptance of this manuscript in this journal in present state.

Response:

Thank you sincerely for your valuable feedback, which has greatly contributed to improving our manuscript. We have thoroughly addressed each of your comments and diligently implemented the recommended changes.

Comment:

1. What is the innovation or advancement of this work in contrast to the authors' previous publications (Nat Commun. 2023, 14, 2361) except for two additional targets?

Response:

Our previous publication (Nature Communications, 2023, 14:2361) and this manuscript both employ Lateral Flow Assays (LFA) in the context of point-of-care diagnostics, yet there are notable differences between the two studies, which are as follows:

1) Transition from Static 2D Images to Time Domain Data Integration: While the previous paper was centered around a sample-to-answer platform using static 15-minute 2D images without considering temporal dynamics, this manuscript enhances the deep learning algorithm to include time domain analysis, utilizing continuous image capture. We apply Convolutional Neural Networks (CNN) and Long Short-Term Memory (LSTM) networks to these time-series images, enabling us to predict ongoing reactions within 1-2 minutes and estimate the final assay outcomes before their actual completion. This advancement in integrating time domain data represents a significant innovation in our research approach.

2) Enhancing Early Prediction via Binding Kinetics from Continuous Imaging: Contrary to the previous paper, which focused on colorimetric outcomes at a set duration (15 minutes), this manuscript innovates by predicting final reaction curves and quantities through continuous change observation. This approach, utilizing a time domain-based predictive algorithm, is especially beneficial in rapid diagnostic scenarios, such as myocardial infarction or emergency room pregnancy tests. Responding to the reviewer's insights, we have conducted additional experiments with Troponin I, a cardiac marker, thereby expanding the scope and demonstrating wider applicability of our method.

3) Enhanced Diagnostic Efficiency with Our Algorithm: Our manuscript achieves a significant reduction in diagnostic time, decreasing it from 15 minutes to only 2 minutes, an approximate 87% decrease. This efficiency surpasses the 58% reduction featured in recent publications, such as "Rapid and stain-free quantification of viral plaque via lens-free

holography and deep learning" in Nature Biomedical Engineering (volume 7, pages 1040–1052, 2023). The movement towards shorter assay times using algorithmic approaches is gaining recognition in prominent scientific journals.

Revisions Made in Response to the Reviewer's Comments:

1) **Introduction Part:** The introduction has been amended to emphasize the distinctions from our previous paper. We altered the text from *“Recently, our group proposed deep learning-assisted smartphone-based LFA (SMART^{AI}-LFA) and demonstrated that integrating clinical sample learning and two-step algorithms enables a cradle-free on-site assay with higher accuracy (>98%)³⁸.”* to *“Recently, our group proposed deep learning-assisted smartphone-based LFA (SMARTAI-LFA) and demonstrated that integrating clinical sample learning and two-step algorithms enables a cradle-free on-site assay with higher accuracy (>98%)³⁹. However, the earlier study primarily highlighted the performance of AI-enhanced colorimetric assays and did not specifically address the reduction of assay time using AI.”* (line 90-95)

2) **Demonstrating Shortened Assay Time with Additional Targets:** Experiments were conducted with Troponin I, using 96 data. The results indicate a reduction in assay time from 15 minutes to 2 minutes, maintaining high levels of accuracy (97.9%), sensitivity (96.9%), and specificity (98.4%). These findings have been added to the results, materials and methods, supplementary information, and Figure 4. we added as *“Next, we further validated the performance of the TIMESAVER assay for non-infectious biomarkers, including Troponin I and hCG for ER. Initially focusing on Troponin I, as shown in Figure 4a-c, we acknowledged its clinical relevance above 0.4 ng/ml, following previous research¹⁷. Therefore, we set a cut-off at 0.5 ng/mL and established a five-class multi-classification using recombinant protein, based on LFA manufacturer’s guideline. This involved training with 618 data, validation with 62 data, and testing with 96 data. The results yielded a sensitivity of 96.9%, specificity of 98.4%, and accuracy of 97.9% (Fig. 4a). In Fig. 4b, the AUC value from the ROC curve was 0.99, and the TIMESAVER demonstrated high accuracy within a 2-minute diagnostic timeframe. TIMESAVER showed some false signals at lower concentrations (Fig. 4c), which appear to be more a limitation of the LFA rather than the algorithm. These results confirm the effectiveness of our algorithm in achieving multi-classification within just 2 minutes of testing, underscoring its utility in rapid diagnostic scenario.”* (line 252-264)

3) **Revising four Figures:** most of figures (Figures 2 to 5) have been rearranged to more effectively illustrate the significant differences between this manuscript and our previous work.

-*Revised Fig. 2:* We have made the algorithm related to object finding clearer and modified the schematic in Fig.2a to distinguish it from our previous paper. The revised Fig. 2 now shows object finding for each of the 12 input data images acquired over 2 minutes (at 10-second intervals) through the use of YOLO (You Only Look Once).

-*Revised Fig. 3:* We have moved the influenza data, previously presented in Fig. 4 of the earlier manuscript, to Fig. 3. This figure now combines these data with the existing COVID-19 results to collectively represent the outcomes for Infectious Diseases (Communicable Diseases). See revised manuscript with red colors.

-*Revised Fig. 4:* The updated Fig. 4 illustrates results pertaining to the application in non-infectious biomarkers. For this purpose, we have included new results for Troponin I, known as a cardiac infarction marker. Additionally, results for hCG, intended for use in emergency rooms (ER), are presented to demonstrate the application in rapid diagnostics for non-communicable diseases. We added paragraph as *“Next, we further validated the performance of the TIMESAVER assay for non-infectious biomarkers, including Troponin I and hCG for ER. Initially focusing on Troponin I, as shown in Figure 4a-c, we acknowledged its clinical relevance above 0.4 ng/ml, following previous research¹⁷. Therefore, we set a cut-off at 0.5 ng/mL and established a five-class multi-classification using recombinant protein, based on LFA manufacturer’s guideline. This involved training with 618 data, validation with 62 data, and testing with 96 data. The results yielded a sensitivity of 96.9%, specificity of 98.4%, and accuracy of 97.9% (Fig. 4a). In Fig. 4b, the AUC value from the ROC curve was 0.99, and the TIMESAVER demonstrated high accuracy within a 2-minute diagnostic timeframe. TIMESAVER showed some false signals at lower concentrations (Fig. 4c), which appear to be more a limitation of the LFA rather than the algorithm. These results confirm the effectiveness of our algorithm in achieving multi-classification within just 2 minutes of testing, underscoring its utility in rapid diagnostic scenario.”* (line 252-264)

We also modified hcG data as *“In emergency room settings, rapid diagnosis of hCG is essential, particularly for assessing pregnancy in patients. (Fig. 4d) demonstrates the sensitivity, specificity, and accuracy for hCG detection within 2 minutes, using test data (n=60). The results revealed that the sensitivity, specificity, and accuracy for hCG were 97.5%, 95.0%, and 96.7%, respectively. The AUC value derived from the ROC curve was 0.95 (Fig. 4e), and the confusion matrix (Fig. 4f) suggests effective performance of the classifier, even when applied in a 2-minute assay utilizing the TIMESAVER model.”* (See line 265-271)

-Revised Fig. 5: We moved the hCG data to revised fig. 4, and rearranged the blind test data in revised fig.5. We revised manuscript as “We presented the results of blind tests using images from a 15-minute assay (Fig. 5b) followed by a 2-minute assay (Fig. 5c), involving both untrained individuals and human experts, as well as the TIMESAVER algorithm, which demonstrated a notable reduction in assay time. The 15-minute assay shown in Fig. 5b was conducted following the manufacturer's guidelines for conventional assays. In these 15-minute assay images, untrained individuals reached an accuracy rate of 70.7%, while human experts attained 78.1%. The lower accuracy compared to the manufacturer’s claim of >90% sensitivity and >99% specificity can be attributed to our inclusion of a substantial number of lower titer data. Nevertheless, the TIMESAVER model surpassed both human experts and untrained individuals in performance, achieving a higher accuracy of 80.6% even in a shortened 2-minute assay.” (See line 311-320)

We have added and revised relevant content to *each figure, caption, and the main body* of the manuscript accordingly.

4) **Changing title:** We changed title from *“Transforming Point-of-Care Diagnostics: 1-Minute Assays Enabled by Time-Series Deep Learning“* to *“Rapid Deep Learning-Assisted Predictive Diagnostics for Point-of-Care Testing”*

2. 10 pg/mL of SARS-CoV-2 N-protein, that is, the COVID-19 diagnostic cut-off value (Nat Commun. 2021, 12, 1931; J Clin Microbiol. 2021, 59(10), e01001-21), should be analyzed by the author's method.

Response:

1) The sensitivity of 10 pg/mL for SARS-CoV-2 N-protein, as suggested by the reviewer, is indeed 1000 times greater than traditional immunoassays. It's well-established that lateral flow assays (LFAs) typically have lower sensitivity compared to ELISA. To address this, our research group recently undertook a study titled "PCR-like performance of rapid test with permselective tunable nanotrap" (Nature Communications, 2023, 14:1520), where we applied sample enrichment techniques to enhance the sensitivity of commercial LFAs significantly. Through the integration of these sample enrichment methods with AI-assisted assay technologies, we believe that achieving high-sensitivity diagnostics is feasible, potentially even detecting concentrations as low as those mentioned by the reviewer.

2) Actually, we conducted blind tests under conditions with sensitivity levels significantly lower than those specified by the manufacturer approved by KFDA (or FDA EUA). This approach is depicted in Figure 5 as *"We presented the results of blind tests using images from a 15-minute assay (Fig. 5b) followed by a 2-minute assay (Fig. 5c), involving both untrained individuals and human experts, as well as the TIMESAVER algorithm, which demonstrated a notable reduction in assay time. The 15-minute assay shown in Fig. 5b was conducted following the manufacturer's guidelines for conventional assays. In these 15-minute assay images, untrained individuals reached an accuracy rate of 70.7%, while human experts attained 78.1%. The lower accuracy compared to the manufacturer's claim of >90% sensitivity and >99% specificity can be attributed to our inclusion of a substantial number of lower titer data. Nevertheless, the TIMESAVER model surpassed both human experts and untrained individuals in performance, achieving a higher accuracy of 80.6% even in a shortened 2-minute assay."* (line 311-320). This approach was facilitated by significantly increasing the number of low viral titer samples. The experiments utilized real patient samples, and all samples were classified based on their Ct values, rather than the NP protein concentration.

3. Whether the time-series deep learning algorithm on the LFA platform has the potential to replace PCR.

Response:

Firstly, at the current stage, we anticipate that our algorithm will serve as an auxiliary function to lateral flow assays (LFA), rather than a replacement for PCR in managing pandemic infectious diseases. Recent publications in the New England Journal of Medicine (NEJM 383, e120 (2020)) and the BMJ (BMJ 372, n208 (2021)) have proposed new diagnostic strategies for handling pandemics. The authors argued that the most effective COVID-19 mitigation could be achieved using frequent, low-cost, simple, and rapid tests due to the exponential growth and spread of SARS-CoV-2. Therefore, rapid and frequent testing for COVID-19 is crucial in minimizing virus transmission, particularly before symptom onset and in asymptomatic cases. We foresee that time-series smartphone-AI could support the strategies recommended by NEJM and BMJ in the near future, as it offers increased accuracy and affordability in line with REASSURED criteria. Furthermore, this smartphone AI platform can be applied to semi-quantitative Point-of-Care Testing (POCT). We recognize that this smartphone AI cannot yet replace PCR, as the currently commercialized LFA test kits exhibit limited performance, having lower sensitivity and higher Coefficient of Variation (CV) compared to molecular diagnostics. Our recent paper, "PCR-like Performance of Rapid Test with Permselective Tunable Nanotrap (Nature Communications, 2023, 14:1520)", addresses this issue. If one could enhance the performance of LFA as demonstrated in the Nanotrap paper, the combination of smartphone AI and highly sensitive LFA could facilitate daily monitoring, similar to the recommendations in the two recent papers (NEJM 383, e120 and BMJ 372, n208).

Secondly, TIMESAVER, our current algorithm, could be a powerful solution for non-infectious diseases where rapid diagnosis is essential. As mentioned in our manuscript, it allows for quick diagnostics in emergency situations such as rapid pregnancy testing in ER or fast detection of myocardial infarction markers (newly added). It provides sufficiently rapid diagnosis within the clinical range, and our paper has convincingly demonstrated the possibility of diagnosing clinically meaningful concentration ranges.

4. Utilizing HCG as the detection target cannot highlight the necessity for a 1-minute assay. Authors may evaluate the performance of the AI-LFA for acute disease diagnostics, such as by choosing the biomarkers of myocardial infarction, acute renal failure, or toxic shock syndrome as the detection target.

Response:

1) In response to the reviewer's comments, I have newly added results for Troponin I. As previously mentioned, we rearranged the order of the figures, displaying infectious diseases (communicable diseases) in the *revised Fig. 3*, and in *revised Fig. 4*, we have included new results for Troponin I, thereby demonstrating the application in non-infectious biomarkers. These results, especially the rapid diagnostics using rapid test kits for both Infectious Diseases (Communicable Diseases) and non-infectious biomarkers, I believe, prove that the TIMESAVER algorithm is an extremely useful technology.

2) Reflecting the reviewer's comments, I have also revised the title of the paper. It has been changed from *“Transforming Point-of-Care Diagnostics: 1-Minute Assays Enabled by Time-Series Deep Learning”* to *“Rapid Deep Learning-Assisted Predictive Diagnostics for Point-of-Care Testing”*

3) For more detailed information on Troponin I, we have revised the results section as follows: *“Next, we further validated the performance of the TIMESAVER assay for non-infectious biomarkers, including Troponin I and hCG for ER. Initially focusing on Troponin I, as shown in Figure 4a-c, we acknowledged its clinical relevance above 0.4 ng/ml, following previous research¹⁷. Therefore, we set a cut-off at 0.5 ng/mL and established a five-class multi-classification using recombinant protein, based on LFA manufacturer’s guideline. This involved training with 618 data, validation with 62 data, and testing with 96 data. The results yielded a sensitivity of 96.9%, specificity of 98.4%, and accuracy of 97.9% (Fig. 4a). In Fig. 4b, the AUC value from the ROC curve was 0.99, and the TIMESAVER demonstrated high accuracy within a 2-minute diagnostic timeframe. TIMESAVER showed some false signals at lower concentrations (Fig. 4c), which appear to be more a limitation of the LFA rather than the algorithm. These results confirm the effectiveness of our algorithm in achieving multi-classification within just 2 minutes of testing, underscoring its utility in rapid diagnostic scenarios.”* (line 252-264)

4) We also revised introduction part (or results part) as *“For example, cardiac troponin I, which is highly specific to myocardial tissue and undetectable in healthy individuals, is significantly elevated in patients with myocardial infarction and can remain elevated for up to 10 days post-necrosis. Levels above 0.4 ng/ml indicate a notably higher 42-day mortality risk¹⁷. Particularly for myocardial infarction patients who present to the emergency room, prompt diagnosis and management are crucial. In such critical scenarios, the rapid identification of diseases and conditions exerts a profound impact on patient outcomes.”* (line 62-68)

5. Using “transforming” to promote the significance of deep learning-assisted LFA is overexaggerated. And, the AI-LFA cannot be called a revolutionary POCT. The aid of AI increases the learning cost for general people, whether an irregular photograph affects the accuracy of detection results, such as mobile phone manufacturer, exposure time, focus, and resolution.

Response:

1) In response to your comment regarding the use of *“transforming”* in the title and the potential overstatement of the impact of AI in LFA, we have revised the title to *“Rapid Deep Learning-Assisted Predictive Diagnostics for Point-of-Care Testing”*.

2)We have moderated the language in the manuscript by removing terms like *“pioneering”* and *“revolutionize”* to ensure a more measured and objective tone.

3)Thank you for your comment regarding the concern that *“The aid of AI increases the learning cost for general people, and whether an irregular photograph impacts the accuracy of detection results, factors such as mobile phone manufacturer, exposure time, focus, and resolution.”*

-Indeed, this was an area of concern for us as well. However, our previous research has shown that issues arising from the type of smartphone, manufacturer, camera, and lighting conditions are not significant and can mostly be overcome through a deep learning-assisted assay. This has been demonstrated in our research team's previous paper in Nature

Communications (2023). Please refer to Fig. 4 e-f in the linked paper for more details (<https://www.nature.com/articles/s41467-023-38104-5>).

-Actually, issues like inevitable shadows caused by multi-lighting conditions during photography can be significant hindrances to AI decision-making. To address this, we have implemented a retest option in APP that allows users to retake and submit data under shadow-free conditions. We are currently preparing for KFSA approval and plan to pursue approvals from the European IVDR and the FDA in the near future.

-The reviewer's apprehension about the escalated learning costs associated with AI assistance is understood. However, our experience indicates that the actual costs of learning are not as substantial as one might expect. We firmly believe that the benefits obtained from this investment far outweigh these costs.

6. To solidify the necessity of the AI-LFA strip, authors may develop a multiplexing LFA strip using fluorescent materials with different excitation wavelengths, since the naked eye-based assay has a limited ability to distinguish the gradients of various colors.

Response:

1) Thank you for the insightful comment. We are actively exploring various applications for expanding our work into immunodiagnostic devices. The method mentioned by the reviewer has been part of our considerations. However, after consulting with manufacturers developing fluorescence-based rapid test kits, we understand that current fluorescence readers are designed for scanning at specific time points, not for continuous monitoring. Our research emphasizes reducing assay times through analysis of time-series images and trained data, which requires data from sequential time frames. Thus, applying our concept to a multiplexing LFA strip using fluorescent materials with different excitation wavelengths is currently challenging under these constraints. Future development of devices capable of capturing continuous or video-based fluorescent images could, however, make expansion into fluorescence imaging feasible.

2) Regarding the application in immunodiagnostic equipment, especially those based on binding kinetics like Surface Plasmon Resonance, we believe our approach is well-suited. We have added a section to the discussion part of our manuscript to address this potential.

We revised conclusion parts

from **“3. Time-Domain Decision Framework: Our algorithm hinges on time-domain decisions. We anticipate its seamless application in assays based on binding kinetics, encompassing techniques like Surface Plasmon Resonance (SPR), Isothermal Titration Calorimetry (ITC), Fluorescence Resonance Energy Transfer (FRET), and more. Furthermore, we aim to broaden its application to various sensing platforms such as Nanowire Sensors, Field-Effect Transistor (FET) Sensors, and Digital Immunoassays.”**

to **“3. Time-Domain Decision Framework: At the core of our approach is an algorithm based on time-domain decisions, specifically designed for assays dependent on binding kinetics. This encompasses techniques such as surface plasmon resonance (SPR), isothermal titration calorimetry (ITC), and fluorescence resonance energy transfer (FRET). Our algorithm's versatility enables potential adaptation to a variety of sensing technologies, including nanowire sensors, field-effect transistor (FET) sensors, and digital immunoassays, broadening its utility across diverse biomedical research and clinical environments. Additionally, we are advancing the integration of this framework with sophisticated biosensing methods like electrochemical impedance spectroscopy (EIS) and quartz crystal microbalance (QCM), with the aim of offering comprehensive analytical capabilities across a wide range of biomolecular interactions.”** (line 383~392)

7. Generally speaking, the control line can confirm the strips work correctly and act as an Internal reference signal to improve the detection accuracy. ROI selection (Figure 2b) requires more data support and principal explanation than present results.

Response:

1) The issue mentioned by the reviewer has already been addressed in our previous paper (Nat Commun., 2023, 14, 2361). We have detailed this in the methods, results, and discussion sections of the previous publication. However, we also described this in the methods section, **“As discussed in our previous work³⁹, we have extensively addressed the challenges related to invalid test results and their decision-making processes. For this study, we used single images of**

the control line captured at either one or two minutes. The cropping of the control line region facilitated the determination of the test strip's validity, allowing for decisions to be made on all invalid cases.” (line 543-546) Through this training on invalid cases, we were able to accurately determine the validity of a test based on the presence or absence of the control line.

2) In response to the reviewer’s comment, we have made the following revisions in the manuscript regarding Fig. 2b: We changed the original statement, *“By selecting the ROI, the chances of detecting the precise concentration of the target biomarker or pathogen are improved, enhancing sensitivity and specificity by reducing false-negative and false-positive results³⁸. We explored two approaches for ROI selection in LFA: using the window and test line only. The window approach achieved an ROI selection rate of 92.9% while testing only the test line improved the rate to 95.2%.”* to *“The selection of the Region of Interest (ROI) enhances the accuracy of detecting the specific concentration of the target biomarker or pathogen, thereby increasing sensitivity and specificity and minimizing the occurrence of false negatives and false positives³⁹. As detailed in our previous research, we investigated two methods for ROI selection in LFAs: focusing on the window and the test line exclusively. The approach centered on the window area achieved a prediction accuracy of 92.9%, while a focus exclusively on the test line enhanced the prediction accuracy to 95.2%.”* (line 148-155)

8. In the test strip, the control line confirms that the test paper is working correctly, while the test line shows whether the samples contain the target. This paper only obtains the diagnostic results through the test line. How do we exclude the wrong results caused by the failure of the test strips (invalid state)?

Response:

We added sentences in method section as *“As discussed in our previous work³⁹, we have extensively addressed the challenges related to invalid test results and their decision-making processes. For this study, we used single images of the control line captured at either one or two minutes. The cropping of the control line region facilitated the determination of the test strip's validity, allowing for decisions to be made on all invalid cases.”* (line 543-546)

9. Very little data is obtained through a 1-minute analysis in this work. Is it appropriate to use the word 1-minute analysis in the title?

Response:

Reflecting the reviewer's comments, I have also revised the title of the paper. It has been changed from "*Transforming Point-of-Care Diagnostics: 1-Minute Assays Enabled by Time-Series Deep Learning*" to "*Rapid Deep Learning-Assisted Predictive Diagnostics for Point-of-Care Testing*"

Following the reviewer's comments, other sections of the manuscript were also revised and are highlighted in red for easy identification. Your helpful comments have allowed us to enhance our manuscript significantly. I sincerely appreciate your thorough review and insightful feedback.

Reviewer #2 (Remarks to the Author):

The authors provide a deep learning methodology to generate an algorithm allowing the radical shortening of assay times for lateral flow assays, and present convincing AUC data for three assays demonstrating that the approach competes well with the conventional (end-point) observation for the same assays.

There is value in shorter assay times; furthermore, especially for home tests such as covid tests impatient users commonly, and without AI assistance, interpret assay results prematurely (essentially operating them outside of approved guidelines), with unclear consequences. A device allowing the interpretation much earlier would alleviate such misuse.

1) What is not shown, however, is a comparison of the results obtained with the time saver algorithm compared to say, visually reading the LFAs after a shortened time without the benefit of the timesaver algorithm or a phone imager.

Response:

Thank you for your constructive and valuable feedback.

1) Conducting a blind test with visual observation alone, as suggested by the reviewer, presents practical challenges. This is largely because it requires gathering many individuals to observe a single test kit simultaneously after a two-minute wait, which is logistically complex and demanding. Therefore, in our study, we opted for a blind test using images for COVID-19 clinical data, as depicted in Fig.5. When human interpretations adhered to the manufacturer's standard 15-minute visual assessment guideline, accuracy rates for human experts stood at 78.1%, and for untrained individuals at 70.7%. However, these rates significantly declined to 64.6% for experts and 59.4% for untrained individuals when the assessment time was reduced to 2 minutes.

2) Despite these challenges, our research team (n=3) determined that there was no substantial difference in accuracy between interpretations made from photographs and those from direct visual observation, when used images with 3000x4000 pixels.

3) Notably, accuracy markedly diminishes when based on a single image taken at either 1 or 2 minutes. Our results suggest that high accuracy is achievable primarily through training with continuous images, as employed in the TIMESAVER algorithm. In contrast, human interpretation, relying typically on a single image, leads to significantly lower accuracy.

2) For many LFAs, especially covid antigen LFAs and higher viral loads, test lines appear very quickly, often within a minute or two, and results could be possibly conclusively interpreted just visually, with good AUC compared to running the same sample sets to normal conclusion. I wonder if the authors could comment on that and maybe present data on the specific value of the timesaver algorithm over just visually interpreting the LFAs after a shortened time.

Response:

Thank you for your helpful comment.

1) As the reviewer recognized, medical diagnostics require the detection of low virus titers or concentrations, demanding good limits of detection and sensitivity, along with a broad dynamic range in the clinical range (low to high). As pointed out by the reviewer, rapid diagnosis is possible with higher virus titers in cases like COVID-19. However, the real challenge and crucial aspect of diagnostic devices reside in detecting lower titers, a critical factor for ensuring accurate diagnostics in such scenarios.

2) In *Figure 5f*, we illustrated the results from both human evaluators (experts and untrained individuals) and the TIMESAVER algorithm, using a heat map based on different virus titers from COVID-19 clinical samples. These samples were categorized into high, middle, middle-low, low, and negative classes according to the manufacturer's color chart. As previously noted, our research team (n=3) determined that there was negligible difference in accuracy

between human interpretations from photographs and direct visual observations. The reviewer correctly points out that human accuracy is relatively high for higher concentrations within a brief 2-minute period. However, as the concentration diminishes, AI's accuracy substantially exceeds that of human observers, a consistency observed in both direct and image-based human observations. These observations are detailed in Figure 5f as follows. *“The heat map indicates that human visual assessment, conducted by both untrained individuals and experts, shows a decrease in accuracy, particularly within the mid-low titer ranges (Fig. 5f). In the mid-low titer category, untrained individuals managed an average accuracy of only 29.2%, while human experts fared slightly better at 37.2%. In contrast, our algorithm achieved an impressive accuracy rate of 84.6%. For the low titer category, the accuracy was even lower, with untrained individuals at 2.8% and human experts at 5.4%, but our deep learning algorithm significantly outperformed at 38.5% accuracy. In cases of high and middle titer concentrations, the TIMESAVER algorithm consistently provided reliable and accurate data, effectively eliminating the variability seen in human visual assessments.”* (line 354-362)

3) In addition, enhance our manuscript, we have made additional revisions to the manuscript.

(1) **Changing title:** We changed title from *“Transforming Point-of-Care Diagnostics: 1-Minute Assays Enabled by Time-Series Deep Learning”* to *“Rapid Deep Learning-Assisted Predictive Diagnostics for Point-of-Care Testing”*

(2) **Introduction Part:** We have revised the introduction to highlight the distinctions from our previous work. The text was changed from *“Recently, our group proposed deep learning-assisted smartphone-based LFA (SMART^{AI}-LFA) and demonstrated that integrating clinical sample learning and two-step algorithms enables a cradle-free on-site assay with higher accuracy (>98%)³⁸.”* to *“Recently, our group proposed deep learning-assisted smartphone-based LFA (SMARTAI-LFA) and demonstrated that integrating clinical sample learning and two-step algorithms enables a cradle-free on-site assay with higher accuracy (>98%)³⁹. However, the earlier study primarily highlighted the performance of AI-enhanced colorimetric assays and did not specifically address the reduction of assay time using AI.”* (line 90-95)

(3) **Demonstrating Shortened Assay Time with Additional Targets:** We conducted experiments with Troponin I using 96 data, showing a reduction in assay time from 15 minutes to 2 minutes while maintaining high accuracy (97.9%), sensitivity (96.9%), and specificity (98.4%). *These results are now included in the materials and methods, supplementary information, and Figure 4.*

(4) **Revising of Four Figures:** We rearranged *most figures (Figures 2 to 5)* to better illustrate the significant differences between this manuscript and our previous work.

Revised Fig. 2: The algorithm related to object finding was clarified, and the schematic in Fig.2a was modified to distinguish it from our previous paper. The revised Fig. 2 now shows object finding for each of the 12 input data images acquired over 2 minutes (at 10-second intervals) using YOLO (You Only Look Once).

Revised Fig. 3: Influenza data from Fig. 4 of the earlier manuscript was moved to Fig. 3, combining it with existing COVID-19 results to represent outcomes for Infectious Diseases (Communicable Diseases).

Revised Fig. 4: The updated Fig. 4 displays results related to non-infectious biomarker, including new findings for Troponin I and results for hCG, demonstrating their application in rapid diagnostics.

We have added relevant content to each figure, caption, and the manuscript's main body.

(5) **Conclusion Part Revised:** from “**3. Time-Domain Decision Framework:** *Our algorithm hinges on time-domain decisions. We anticipate its seamless application in assays based on binding kinetics, encompassing techniques like Surface Plasmon Resonance (SPR), Isothermal Titration Calorimetry (ITC), Fluorescence Resonance Energy Transfer (FRET), and more. Furthermore, we aim to broaden its application to various sensing platforms such as Nanowire Sensors, Field-Effect Transistor (FET) Sensors, and Digital Immunoassays.*” to “**3. Time-Domain Decision Framework:** *At the core of our approach is an algorithm based on time-domain decisions, specifically designed for assays dependent on binding kinetics. This encompasses techniques such as surface plasmon resonance (SPR), isothermal titration calorimetry (ITC), and fluorescence resonance energy transfer*

(FRET). Our algorithm's versatility enables potential adaptation to a variety of sensing technologies, including nanowire sensors, field-effect transistor (FET) sensors, and digital immunoassays, broadening its utility across diverse biomedical research and clinical environments. Additionally, we are advancing the integration of this framework with sophisticated biosensing methods like electrochemical impedance spectroscopy (EIS) and quartz crystal microbalance (QCM), with the aim of offering comprehensive analytical capabilities across a wide range of biomolecular interactions..”
(line 383-392)

Following the reviewer's comments, other sections of the manuscript were also revised and are highlighted in red for easy identification. Your helpful comments have allowed us to enhance our manuscript significantly. I sincerely appreciate your thorough review and insightful feedback.

REVIEWERS' COMMENTS

Reviewer #1 (Remarks to the Author):

I have thoroughly reviewed the manuscript, along with the author's responses to feedback from other reviewers. I am pleased to note that the manuscript has undergone substantial improvements, and the concerns I previously had regarding its significance and novelty have been effectively addressed in the revised draft. Given these improvements, I wholeheartedly recommend this article for publication in its current form.

Reviewer #2 (Remarks to the Author):

The authors thoroughly and appropriately addressed the reviewer comments.